# Predictive Uncertainty Estimation via Prior Networks

**Andrey Malinin**
Department of Engineering
University of Cambridge
am969@cam.ac.uk

**Mark Gales**
Department of Engineering
University of Cambridge
mjfg@eng.cam.ac.uk

## Abstract

Estimating how uncertain an AI system is in its predictions is important to improve the safety of such systems. Uncertainty in predictive can result from uncertainty in model parameters, irreducible *data uncertainty* and uncertainty due to distributional mismatch between the test and training data distributions. Different actions might be taken depending on the source of the uncertainty so it is important to be able to distinguish between them. Recently, baseline tasks and metrics have been defined and several practical methods to estimate uncertainty developed. These methods, however, attempt to model uncertainty due to distributional mismatch either implicitly through *model uncertainty* or as *data uncertainty*. This work proposes a new framework for modeling predictive uncertainty called Prior Networks (PNs) which explicitly models *distributional uncertainty*. PNs do this by parameterizing a prior distribution over predictive distributions. This work focuses on uncertainty for classification and evaluates PNs on the tasks of identifying out-of-distribution (OOD) samples and detecting misclassification on the MNIST and CIFAR-10 datasets, where they are found to outperform previous methods. Experiments on synthetic and MNIST and CIFAR-10 data show that unlike previous non-Bayesian methods PNs are able to distinguish between data and distributional uncertainty.

## 1 Introduction

Neural Networks (NNs) have become the dominant approach to addressing computer vision (CV) [1, 2, 3], natural language processing (NLP) [4, 5, 6], speech recognition (ASR) [7, 8] and bio-informatics (BI) [9, 10] tasks. Despite impressive, and ever improving, supervised learning performance, NNs tend to make over-confident predictions [11] and until recently have been unable to provide measures of uncertainty in their predictions. Estimating uncertainty in a model's predictions is important, as it enables, for example, the safety of an AI system [12] to be increased by acting on the model's prediction in an informed manner. This is crucial to applications where the cost of an error is high, such as in autonomous vehicle control and medical, financial and legal fields.

Recently notable progress has been made on predictive uncertainty for Deep Learning through the definition of baselines, tasks and metrics [13] and the development of practical methods for estimating uncertainty. One class of approaches stems from Bayesian Neural Networks [14, 15, 16, 17]. Traditionally, these approaches have been computationally more demanding and conceptually more complicated than non-Bayesian NNs. Crucially, their performance depends on the form of approximation made due to computational constraints and the nature of the prior distribution over parameters. A recent development has been the technique of Monte-Carlo Dropout [18], which estimates predictive uncertainty using an ensemble of multiple stochastic forward passes and computing the mean and spread of the ensemble. This technique has been successfully applied to tasks in computer vision [19, 20]. A number of non-Bayesian ensemble approaches have also been proposed. One approach based on explicitly training an ensemble of DNNs, called Deep Ensembles [11], yields competitive uncertainty estimates to MC dropout. Another class of approaches, developed

for both regression [21] and classification [22], involves explicitly training a model in a multi-task fashion to minimize its Kullback-Leibler (KL) divergence to both a sharp in-domain predictive posterior and a flat out-of-domain predictive posterior, where the out-of-domain inputs are sampled either from a synthetic noise distribution or a different dataset during training. These methods are explicitly trained to detect out-of-distribution inputs and have the advantage of being more computationally efficient at test time.

The primary issue with these approaches is that they conflate different aspects of predictive uncertainty, which results from three separate factors - *model uncertainty*, *data uncertainty* and *distributional uncertainty*. *Model uncertainty*, or *epistemic uncertainty* [23], measures the uncertainty in estimating the model parameters given the training data - this measures how well the model is matched to the data. *Model uncertainty* is reducible[1] as the size of training data increases. *Data uncertainty*, or *aleatoric uncertainty* [23], is irreducible uncertainty which arises from the natural complexity of the data, such as class overlap, label noise, homoscedastic and heteroscedastic noise. *Data uncertainty* can be considered a 'known-unknown' - the model understands (knows) the data and can confidently state whether a given input is difficult to classify (an unknown). *Distributional uncertainty* arises due to mismatch between the training and test distributions (also called dataset shift [24]) - a situation which often arises for real world problems. *Distributional uncertainty* is an 'unknown-unknown' - the model is unfamiliar with the test data and thus cannot confidently make predictions. The approaches discussed above either conflate *distributional uncertainty* with *data uncertainty* or implicitly model *distributional uncertainty* through *model uncertainty*, as in Bayesian approaches. The ability to separately model the 3 types of predictive uncertainty is important, as different actions can be taken by the model depending on the source of uncertainty. For example, in active learning tasks detection of *distributional uncertainty* would indicate the need to collect training data from this distribution. This work addresses the explicit prediction of each of the three types of predictive uncertainty by extending the work done in [21, 22] while taking inspiration from Bayesian approaches.

**Summary of Contributions**. This work describes the limitations of previous methods of obtaining uncertainty estimates and proposes a new framework for modeling predictive uncertainty, called Prior Networks (PNs), which allows *distributional uncertainty* to be treated as distinct from both *data uncertainty* and *model uncertainty*. This work focuses on the application of PNs to classification tasks. Additionally, this work presents a discussion of a range of uncertainty metrics in the context of each source of uncertainty. Experiments on synthetic and real data show that unlike previous non-Bayesian methods PNs are able to distinguish between *data uncertainty* and *distributional uncertainty*. Finally, PNs are evaluated[2] on the tasks of identifying out-of-distribution (OOD) samples and detecting misclassification outlined in [13], where they outperform previous methods on the MNIST and CIFAR-10 datasets.

## 2   Current Approaches to Uncertainty Estimation

This section describes current approaches to predictive uncertainty estimation. Consider a distribution $\mathrm{p}(\boldsymbol{x}, y)$ over input features $\boldsymbol{x}$ and labels $y$. For image classification $\boldsymbol{x}$ corresponds to images and $y$ object labels. In a Bayesian framework the predictive uncertainty of a classification model $\mathrm{P}(\omega_c | \boldsymbol{x}^*, \mathcal{D})$[3] trained on a finite dataset $\mathcal{D} = \{\boldsymbol{x}_j, y_j\}_{j=1}^N \sim \mathrm{p}(\boldsymbol{x}, y)$ will result from *data (aleatoric) uncertainty* and *model (epistemic) uncertainty*. A model's estimates of *data uncertainty* are described by the posterior distribution over class labels given a set of model parameters $\boldsymbol{\theta}$ and *model uncertainty* is described by the posterior distribution over the parameters given the data (eq. 1).

$$\mathrm{P}(\omega_c | \boldsymbol{x}^*, \mathcal{D}) = \int \underbrace{\mathrm{P}(\omega_c | \boldsymbol{x}^*, \boldsymbol{\theta})}_{Data} \underbrace{\mathrm{p}(\boldsymbol{\theta} | \mathcal{D})}_{Model} d\boldsymbol{\theta} \tag{1}$$

Here, uncertainty in the model parameters induces a distribution over distributions $\mathrm{P}(\omega_c | \boldsymbol{x}^*, \boldsymbol{\theta})$. The expected distribution $\mathrm{P}(\omega_c | \boldsymbol{x}^*, \mathcal{D})$ is obtained by marginalizing out the parameters $\boldsymbol{\theta}$. Unfortunately, obtaining the true posterior $\mathrm{p}(\boldsymbol{\theta} | \mathcal{D})$ using Bayes' rule is intractable, and it is necessary to use either an explicit or implicit variational approximation $\mathrm{q}(\boldsymbol{\theta})$ [25, 26, 27, 28]:

$$\mathrm{p}(\boldsymbol{\theta} | \mathcal{D}) \approx \mathrm{q}(\boldsymbol{\theta}) \tag{2}$$

Furthermore, the integral in eq. 1 is also intractable for neural networks and is typically approximated via sampling (eq. 3), using approaches like Monte-Carlo dropout [18], Langevin Dynamics [29] or explicit ensembling [11]. Thus,

$$\mathrm{P}(\omega_c|\boldsymbol{x}^*, \mathcal{D}) \approx \frac{1}{M} \sum_{i=1}^{M} \mathrm{P}(\omega_c|\boldsymbol{x}^*, \boldsymbol{\theta}^{(i)}), \ \boldsymbol{\theta}^{(i)} \sim \mathsf{q}(\boldsymbol{\theta}) \tag{3}$$

Each $\mathrm{P}(\omega_c|\boldsymbol{x}^*, \boldsymbol{\theta}^{(i)})$ in an ensemble $\{\mathrm{P}(\omega_c|\boldsymbol{x}^*, \boldsymbol{\theta}^{(i)})\}_{i=1}^{M}$ obtained sampled from $\mathsf{q}(\boldsymbol{\theta})$ is a categorical distribution $\boldsymbol{\mu}$ [4] over class labels $y$ conditioned on the input $\boldsymbol{x}^*$, and can be visualized as a point on a simplex. For the same $\boldsymbol{x}^*$ this ensemble is a collection of points on a simplex (fig. 1a), which can be seen as samples of categorical distributions from an *implicit* conditional distribution over a simplex (fig. 1b) induced via the posterior over model parameters.

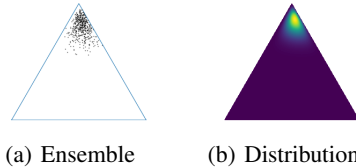

(a) Ensemble      (b) Distribution

Figure 1: Distributions on a Simplex

By selecting an appropriate approximate inference scheme and model prior $\mathsf{p}(\boldsymbol{\theta})$ Bayesian approaches aim to craft an approximate model posterior $\mathsf{q}(\boldsymbol{\theta})$ such that the ensemble $\{\mathrm{P}(\omega_c|\boldsymbol{x}^*, \boldsymbol{\theta}^{(i)})\}_{i=1}^{M}$ is consistent in the region of training data, and becomes increasingly diverse when the input $\boldsymbol{x}^*$ is far from the training data. Thus, these approaches aim to craft an implicit conditional distribution over a simplex (fig. 1b) with the attributes that it is sharp at the corners of a simplex for inputs similar to the training data and flat over the simplex for out-of-distribution inputs. Given an ensemble from such a distribution, the entropy of the expected distribution $\mathrm{P}(\omega_c|\boldsymbol{x}^*, \mathcal{D})$ will indicate uncertainty in predictions. It is not possible, however, to determine from the entropy whether this uncertainty is due to a high degree of *data uncertainty*, or whether the input is far from the region of training data. It is necessary to use measures of spread of the ensemble, such as Mutual Information, to assess uncertainty in predictions due to *model uncertainty*. This allows sources of uncertainty to be determined.

In practice, however, for deep, distributed black-box models with tens of millions of parameters, such as DNNs, it is difficult to select an appropriate model prior and approximate inference scheme to craft a model posterior which induces an implicit distribution with the desired properties. This makes it hard to guarantee the desired properties of the induced distribution for current state-of-the-art Deep Learning approaches. Furthermore, creating an ensemble can be computationally expensive.

An alternative, non-Bayesian class of approaches derives measures of uncertainty via the predictive posteriors of regression [21] and classification [13, 22, 30] DNNs. Here, DNNs are explicitly trained [22, 21] to yield high entropy posterior distributions for out-of-distribution inputs. These approaches are easy to train and inference is computationally cheap. However, a high entropy posterior over classes could indicate uncertainty in the prediction due to *either* an in-distribution input in a region of class overlap or an out-of-distribution input far from the training data. Thus, it is not possible to robustly determine the source of uncertainty using these approaches. Further discussion of uncertainty measures can be found in section 4.

## 3 Prior Networks

Having described existing approaches, an alternative approach to modeling predictive uncertainty, called Prior Networks, is proposed in this section. As previously described, Bayesian approaches aim to construct an implicit conditional distribution over distributions on a simplex (fig 1b) with certain desirable attributes by appropriate selection of model prior and approximate inference method. In practice this is a difficult task and an open research problem.

This work proposes to instead *explicitly* parameterize a distribution over distributions on a simplex, $p(\boldsymbol{\mu}|\boldsymbol{x}^*,\boldsymbol{\theta})$, using a DNN referred to as a *Prior Network* and train it to behave like the implicit distribution in the Bayesian approach. Specifically, when it is confident in its prediction a Prior Network should yield a sharp distribution centered on one of the corners of the simplex (fig. 2a). For an input in a region with high degrees of noise or class overlap (*data uncertainty*) a Prior Network should yield a sharp distribution focused on the center of the simplex, which corresponds to being confident in predicting a flat categorical distribution over class labels (known-unknown) (fig. 2b). Finally, for 'out-of-distribution' inputs the Prior Network should yield a flat distribution over the simplex, indicating large uncertainty in the mapping $\boldsymbol{x} \mapsto y$ (unknown-unknown) (fig. 2c).

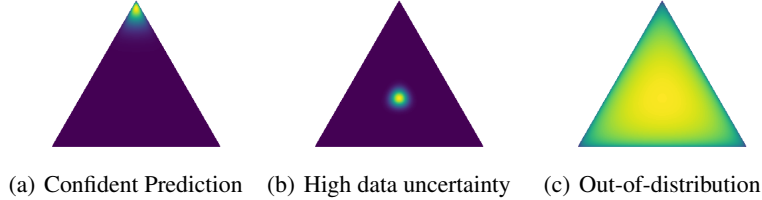

(a) Confident Prediction    (b) High data uncertainty    (c) Out-of-distribution

Figure 2: Desired behaviors of a distribution over distributions

In the Bayesian framework *distributional uncertainty*, or uncertainty due to mismatch between the distributions of test and training data, is considered a part of *model uncertainty*. In this work it will be considered to be a source of uncertainty separate from *data uncertainty* or *model uncertainty*. Prior Networks will be explicitly constructed to capture *data uncertainty* and *distributional uncertainty*. In Prior Networks *data uncertainty* is described by the point-estimate categorical distribution $\boldsymbol{\mu}$ and *distributional uncertainty* is described by the distribution over predictive categoricals $p(\boldsymbol{\mu}|\boldsymbol{x}^*,\boldsymbol{\theta})$. The parameters $\boldsymbol{\theta}$ of the Prior Network must encapsulate knowledge both about the in-domain distribution and the decision boundary which separates the in-domain region from everything else. Construction of a Prior Network is discussed in sections 3.1 and 3.2. Before this it is necessary to discuss its theoretical properties.

Consider modifying eq. 1 by introducing the term $p(\boldsymbol{\mu}|\boldsymbol{x}^*,\boldsymbol{\theta})$ as follows:

$$P(\omega_c|\boldsymbol{x}^*,\mathcal{D}) = \int \int \underbrace{p(\omega_c|\boldsymbol{\mu})}_{Data} \underbrace{p(\boldsymbol{\mu}|\boldsymbol{x}^*,\boldsymbol{\theta})}_{Distributional} \underbrace{p(\boldsymbol{\theta}|\mathcal{D})}_{Model} d\boldsymbol{\mu}d\boldsymbol{\theta} \tag{4}$$

In this expression *data, distribution* and *model uncertainty* are now each modeled by a separate term within an interpretable probabilistic framework. The relationship between uncertainties is made explicit - *model uncertainty* affects estimates of *distributional uncertainty*, which in turn affects the estimates of *data uncertainty*. This is expected, as a large degree of *model uncertainty* will yield a large variation in $p(\boldsymbol{\mu}|\boldsymbol{x}^*,\boldsymbol{\theta})$, and large uncertainty in $\boldsymbol{\mu}$ will lead to a large uncertainty in estimates of *data uncertainty*. Thus, *model uncertainty* affects estimates of *data* and *distributional uncertainties*, and *distributional uncertainty* affects estimates of *data uncertainty*. This forms a hierarchical model - there are now three layers of uncertainty: the posterior over classes, the per-data prior distribution and the global posterior distribution over model parameters. Similar constructions have been previously explored for non-neural Bayesian models, such as Latent Dirichlet Allocation [31]. However, typically additional levels of uncertainty are added in order to increase the flexibility of models, and predictions are obtained by marginalizing or sampling. In this work, however, the additional level of uncertainty is added in order to be able to extract additional measures of uncertainty, depending on how the model is marginalized. For example, consider marginalizing out $\boldsymbol{\mu}$ in eq. 4, thus re-obtaining eq. 1:

$$\int \Big[ \int p(\omega_c|\boldsymbol{\mu})p(\boldsymbol{\mu}|\boldsymbol{x}^*,\boldsymbol{\theta})d\boldsymbol{\mu} \Big] p(\boldsymbol{\theta}|\mathcal{D})d\boldsymbol{\theta} = \int P(\omega_c|\boldsymbol{x}^*,\boldsymbol{\theta})p(\boldsymbol{\theta}|\mathcal{D})d\boldsymbol{\theta} \tag{5}$$

Since the distribution over $\boldsymbol{\mu}$ is lost in the marginalization it is unknown how sharp or flat it was around the point estimate. If the expected categorical $P(\omega_c|\boldsymbol{x}^*,\boldsymbol{\theta})$ is "flat" it is now unknown whether this is due to high data or *distributional uncertainty*. In this situation, it will be necessary to again rely on measures which assess the spread of an MC ensemble, like mutual information (section 4), to establish the source of uncertainty. Thus, Prior Networks are consistent with previous approaches to

modeling uncertainty, both Bayesian and non-Bayesian - they can be viewed as an 'extra tool in the uncertainty toolbox' which is explicitly crafted to capture the effects of distributional mismatch in a probabilistically interpretable way. Alternatively, consider marginalizing out $\boldsymbol{\theta}$ in eq. 4 as follows:

$$\int \mathrm{p}(\omega_c|\boldsymbol{\mu})\Big[\int \mathrm{p}(\boldsymbol{\mu}|\boldsymbol{x}^*,\boldsymbol{\theta})\mathrm{p}(\boldsymbol{\theta}|\mathcal{D})d\boldsymbol{\theta}\Big]d\boldsymbol{\mu} = \int \mathrm{p}(\omega_c|\boldsymbol{\mu})\mathrm{p}(\boldsymbol{\mu}|\boldsymbol{x}^*,\mathcal{D})d\boldsymbol{\mu} \qquad (6)$$

This yields expected estimates of *data* and *distributional uncertainty* given *model uncertainty*. Eq. 6 can be seen as a modification of eq. 1 where the model is redefined as $\mathrm{p}(\omega_c|\boldsymbol{\mu})$ and the distribution over model parameters $\mathrm{p}(\boldsymbol{\mu}|\boldsymbol{x}^*,\mathcal{D})$ is now conditional on both the training data $\mathcal{D}$ and the test input $\boldsymbol{x}^*$. This explicitly yields the distribution over the simplex which the Bayesian approach implicitly induces. Further discussion of how measures of uncertainty are derived from the marginalizations of equation 4 is presented in section 4.

Unfortunately, like eq. 1, the marginalization in eq. 6 is generally intractable, though it can be approximated via Bayesian MC methods. For simplicity, this work will assume that a point-estimate (eq. 7) of the parameters will be sufficient given appropriate regularization and training data size.

$$\mathrm{p}(\boldsymbol{\theta}|\mathcal{D}) = \delta(\boldsymbol{\theta} - \hat{\boldsymbol{\theta}}) \implies \mathrm{p}(\boldsymbol{\mu}|\boldsymbol{x}^*;\mathcal{D}) \approx \mathrm{p}(\boldsymbol{\mu}|\boldsymbol{x}^*;\hat{\boldsymbol{\theta}}) \qquad (7)$$

### 3.1 Dirichlet Prior Networks

A Prior Network for classification parametrizes a distribution over a simplex, such as a Dirichlet (eq. 8), Mixture of Dirichlet distributions or the Logistic-Normal distribution. In this work the Dirichlet distribution is chosen due to its tractable analytic properties. A Dirichlet distribution is a prior distribution over categorical distribution, which is parameterized by its concentration parameters $\boldsymbol{\alpha}$, where $\alpha_0$, the sum of all $\alpha_c$, is called the *precision* of the Dirichlet distribution. Higher values of $\alpha_0$ lead to sharper distributions.

$$\mathrm{Dir}(\boldsymbol{\mu}|\boldsymbol{\alpha}) = \frac{\Gamma(\alpha_0)}{\prod_{c=1}^{K}\Gamma(\alpha_c)}\prod_{c=1}^{K}\mu_c^{\alpha_c-1}, \quad \alpha_c > 0, \; \alpha_0 = \sum_{c=1}^{K}\alpha_c \qquad (8)$$

A Prior Network which parametrizes a Dirichlet will be referred to as a *Dirichlet Prior Network* (DPN). A DPN will generate the concentration parameters $\boldsymbol{\alpha}$ of the Dirichlet distribution.

$$\mathrm{p}(\boldsymbol{\mu}|\boldsymbol{x}^*;\hat{\boldsymbol{\theta}}) = \mathrm{Dir}(\boldsymbol{\mu}|\boldsymbol{\alpha}), \quad \boldsymbol{\alpha} = \boldsymbol{f}(\boldsymbol{x}^*;\hat{\boldsymbol{\theta}}) \qquad (9)$$

The posterior over class labels will be given by the mean of the Dirichlet:

$$\mathrm{P}(\omega_c|\boldsymbol{x}^*;\hat{\boldsymbol{\theta}}) = \int \mathrm{p}(\omega_c|\boldsymbol{\mu})\mathrm{p}(\boldsymbol{\mu}|\boldsymbol{x}^*;\hat{\boldsymbol{\theta}})d\boldsymbol{\mu} = \frac{\alpha_c}{\alpha_0} \qquad (10)$$

If an exponential output function is used for the DPN, where $\alpha_c = e^{z_c}$, then the expected posterior probability of a label $\omega_c$ is given by the output of the softmax (eq. 11).

$$\mathrm{P}(\omega_c|\boldsymbol{x}^*;\hat{\boldsymbol{\theta}}) = \frac{e^{z_c(\boldsymbol{x}^*)}}{\sum_{k=1}^{K}e^{z_k(\boldsymbol{x}^*)}} \qquad (11)$$

Thus, standard DNNs for classification with a softmax output function can be viewed as predicting the expected categorical distribution under a Dirichlet prior. The mean, however, is insensitive to arbitrary scaling of $\alpha_c$. Thus the precision $\alpha_0$, which controls the sharpness of the Dirichlet, is degenerate under standard cross-entropy training. It is necessary to change the cost function to explicitly train a DPN to yield a sharp or flat prior distribution around the expected categorical depending on the input data.

### 3.2 Dirichlet Prior Network Training

There are potentially many ways in which a Prior Network can be trained and it is not the focus of this work to investigate them all. This work considers one approach to training a DPN based on the work done in [21, 22] and here. The DPN is *explicitly* trained in a multi-task fashion to minimize the KL divergence (eq. 12) between the model and a sharp Dirichlet distribution focused on the appropriate class for in-distribution data, and between the model and a flat Dirichlet distribution for

out-of-distribution data. A flat Dirichlet is chosen as the uncertain distribution in accordance with the principle of insufficient reason [32], as all possible categorical distributions are equiprobable.

$$\mathcal{L}(\boldsymbol{\theta}) = \mathbb{E}_{\mathtt{p_{in}}(\boldsymbol{x})}[KL[\mathtt{Dir}(\boldsymbol{\mu}|\hat{\boldsymbol{\alpha}})||\mathtt{p}(\boldsymbol{\mu}|\boldsymbol{x};\boldsymbol{\theta})]] + \mathbb{E}_{\mathtt{p_{out}}(\boldsymbol{x})}[KL[\mathtt{Dir}(\boldsymbol{\mu}|\tilde{\boldsymbol{\alpha}})||\mathtt{p}(\boldsymbol{\mu}|\boldsymbol{x};\boldsymbol{\theta})]] \qquad (12)$$

In order to train using this loss function the in-distribution targets $\hat{\boldsymbol{\alpha}}$ and out-of-distribution targets $\tilde{\boldsymbol{\alpha}}$ must be defined. It is simple to specify a flat Dirichlet distribution by setting all $\tilde{\alpha}_c = 1$. However, directly setting the in-distribution target $\hat{\alpha}_c$ is not convenient. Instead the concentration parameters $\hat{\alpha}_c$ are re-parametrized into $\hat{\alpha}_0$, the target precision, and the means $\hat{\mu}_c = \frac{\hat{\alpha}_c}{\hat{\alpha}_0}$. $\hat{\alpha}_0$ is a hyper-parameter set during training and the means are simply the 1-hot targets used for classification. A further complication is that learning sparse '1-hot' continuous distributions, which are effectively delta functions, is challenging under the defined KL loss, as the error surface becomes poorly suited for optimization. There are two solutions - first, it is possible to smooth the target means (eq. 13), which redistributes a small amount of probability density to the other corners of the Dirichlet. Alternatively, teacher-student training [33] can be used to specify non-sparse target means $\hat{\boldsymbol{\mu}}$. The smoothing approach is used in this work. Additionally, cross-entropy can be used as an auxiliary loss for in-distribution data.

$$\hat{\mu}_c = \left\{ \begin{array}{ll} 1 - (K-1)\epsilon & if \ \delta(y = \omega_c) = 1 \\ \epsilon & if \ \delta(y = \omega_c) = 0 \end{array} \right. \qquad (13)$$

The multi-task training objective (eq. 12) requires samples of $\tilde{\boldsymbol{x}}$ from the out-of-domain distribution $\mathtt{p_{out}}(\boldsymbol{x})$. However, the true out-of-domain distribution is unknown and samples are unavailable. One solution is to synthetically generate points on the boundary of the in-domain region using a generative model [21, 22]. An alternative is to use a different, real dataset as a set of samples from the out-of-domain distribution [22].

## 4 Uncertainty Measures

The previous section introduced a new framework for modeling uncertainty. This section explores a range of measures for quantifying uncertainty given a trained DNN, DPN or Bayesian MC ensemble. The discussion is broken down into 4 classes of measure, depending on how eq. 4 is marginalized. Details of derivation can be found in Appendix C.

The first class derives measures of uncertainty from the expected predictive categorical $\mathtt{P}(\omega_c|\boldsymbol{x}^*;\mathcal{D})$, given a full marginalization of eq. 4 which can be approximated either with a point estimate of the parameters $\hat{\boldsymbol{\theta}}$ or a Bayesian MC ensemble. The first measure is the probability of the predicted class (mode), or *max probability* (eq. 14), which is a measure of confidence in the prediction used in [13, 22, 30, 23, 11].

$$\mathcal{P} = \max_c \mathtt{P}(\omega_c|\boldsymbol{x}^*;\mathcal{D}) \qquad (14)$$

The second measure is the *entropy* (eq. 15) of the predictive distribution [23, 18, 11]. It behaves similar to max probability, but represents the uncertainty encapsulated in the entire distribution.

$$\mathcal{H}[\mathtt{P}(y|\boldsymbol{x}^*;\mathcal{D})] = -\sum_{c=1}^{K} \mathtt{P}(\omega_c|\boldsymbol{x}^*;\mathcal{D}) \ln(\mathtt{P}(\omega_c|\boldsymbol{x}^*;\mathcal{D})) \qquad (15)$$

Max probability and entropy of the expected distribution can be seen as measures of the *total uncertainty* in predictions.

The second class of measures considers marginalizing out $\boldsymbol{\mu}$ in eq. 4, yielding eq. 1. *Mutual Information* (MI) [23] between the categorical label $y$ and the parameters of the model $\boldsymbol{\theta}$ is a measure of the spread of an ensemble $\{\mathtt{P}(\omega_c|\boldsymbol{x}^*,\boldsymbol{\theta}^{(i)})\}_{i=1}^{M}$ [18] which assess uncertainty in predictions due to *model uncertainty*. Thus, MI implicitly captures elements of distributional uncertainty. MI can be expressed as the difference of the total uncertainty, captured by the entropy of expected distribution, and the expected data uncertainty, captured by expected entropy of each member of the ensemble (eq. 16). This interpretation was given in [34].

$$\underbrace{\mathcal{I}[y, \boldsymbol{\theta}|\boldsymbol{x}^*, \mathcal{D}]}_{Model \ Uncertainty} = \underbrace{\mathcal{H}[\mathbb{E}_{\mathtt{p}(\boldsymbol{\theta}|\mathcal{D})}[\mathtt{P}(y|\boldsymbol{x}^*,\boldsymbol{\theta})]]}_{Total \ Uncertainty} - \underbrace{\mathbb{E}_{\mathtt{p}(\boldsymbol{\theta}|\mathcal{D})}[\mathcal{H}[\mathtt{P}(y|\boldsymbol{x}^*,\boldsymbol{\theta})]]}_{Expected \ Data \ Uncertainty} \qquad (16)$$

The third class of measures considers marginalizing out $\boldsymbol{\theta}$ in eq. 4, yielding eq. 6. The first measure in this class is the mutual information between $y$ and $\boldsymbol{\mu}$ (eq. 17), which behaves in exactly the same way as MI between $y$ and $\boldsymbol{\theta}$, but the spread is now explicitly due to distributional uncertainty, rather than model uncertainty.

$$\underbrace{\mathcal{I}[y, \boldsymbol{\mu}|\boldsymbol{x}^*; \mathcal{D}]}_{Distributional\ Uncertainty} = \underbrace{\mathcal{H}[\mathbb{E}_{\mathrm{p}(\boldsymbol{\mu}|\boldsymbol{x}^*;\mathcal{D})}[\mathrm{P}(y|\boldsymbol{\mu})]]}_{Total\ Uncertainty} - \underbrace{\mathbb{E}_{\mathrm{p}(\boldsymbol{\mu}|\boldsymbol{x}^*;\mathcal{D})}[\mathcal{H}[\mathrm{P}(y|\boldsymbol{\mu})]]}_{Expected\ Data\ Uncertainty} \quad (17)$$

Another measure of uncertainty is the *differential entropy* (eq. 18) of the DPN. This measure is maximized when all categorical distributions are equiprobable, which occurs when the Dirichlet Distribution is flat - in other words when there is the greatest variety of samples from the Dirichlet prior. Differential entropy is well suited to measuring distributional uncertainty, as it can be low even if the expected categorical under the Dirichlet prior has high entropy, and also captures elements of data uncertainty.

$$\mathcal{H}[\mathrm{p}(\boldsymbol{\mu}|\boldsymbol{x}^*; \mathcal{D})] = -\int_{\mathcal{S}^{K-1}} \mathrm{p}(\boldsymbol{\mu}|\boldsymbol{x}^*; \mathcal{D}) \ln(\mathrm{p}(\boldsymbol{\mu}|\boldsymbol{x}^*; \mathcal{D})) d\boldsymbol{\mu} \quad (18)$$

The final class of measures uses the full eq. 4 and assesses the spread of $\mathrm{p}(\boldsymbol{\mu}|\boldsymbol{x}^*; \boldsymbol{\theta})$ due to model uncertainty via the MI between $\boldsymbol{\mu}$ and $\boldsymbol{\theta}$, which can be computed via Bayesian ensemble approaches.

## 5 Experiments

The previous sections discussed modeling different aspects of predictive uncertainty and presented several measures of quantifying it. This section compares the proposed and previous methods in two sets of experiments. The first experiment illustrates the advantages of a DPN over other non-Bayesian methods [22, 30] on synthetic data and the second set of experiments evaluate DPNs on MNIST and CIFAR-10 and compares them to DNNs and ensembles generated via Monte-Carlo Dropout (MCDP) on the tasks of misclassification detection and out-of-distribution data detection. The experimental setup is described in Appendix A and additional experiments are described in Appendix B.

### 5.1 Synthetic Experiments

A synthetic experiment was designed to illustrate the limitation of using uncertainty measures derived from $\mathrm{P}(\omega_c|\boldsymbol{x}^*; \mathcal{D})$ [22, 30] to detect out-of-distribution samples. A simple dataset with 3 Gaussian distributed classes with equidistant means and tied isotropic variance $\sigma$ is created. The classes are non-overlapping when $\sigma = 1$ (fig. 3a) and overlap when $\sigma = 4$ (fig. 3d). The entropy of the *true* posterior over class labels is plotted in blue in figures 3a and 3d, which show that when the classes are distinct the entropy is high only on the decision boundaries, but when the classes overlap the entropy is high also within the data region. A small DPN with 1 hidden layer of 50 neurons is trained on this data. Figures 3b and 3c show that when classes are distinct both the entropy of the DPN's predictive posterior and the differential entropy of the DPN have identical behaviour - low in the region of data and high elsewhere, allowing in-distribution and out-of-distribution regions to be distinguished. Figures 3e and 3f, however, show that when there is a large degree of class overlap the entropy and differential entropy have different behavior - entropy is high both in region of class overlap and far from training data, making difficult to distinguish out-of-distribution samples and in-distribution samples at a decision boundary. In contrast, the differential entropy is low over the whole region of training data and high outside, allowing the in-distribution region to be clearly distinguished from the out-of-distribution region.

### 5.2 MNIST and CIFAR-10 Experiments

An in-domain misclassification detection experiment and an out-of-distribution (OOD) input detection experiment were run on the MNIST and CIFAR-10 datasets [35, 36] to assess the DPN's ability to estimate uncertainty. The misclassification detection experiment involves detecting whether a given prediction is incorrect given an uncertainty measure. Misclassifications are chosen as the positive class. The misclassification detection experiment was run on the MNIST valid+test set and the CIFAR-10 test set. The out-of-distribution detection experiment involves detecting whether an input

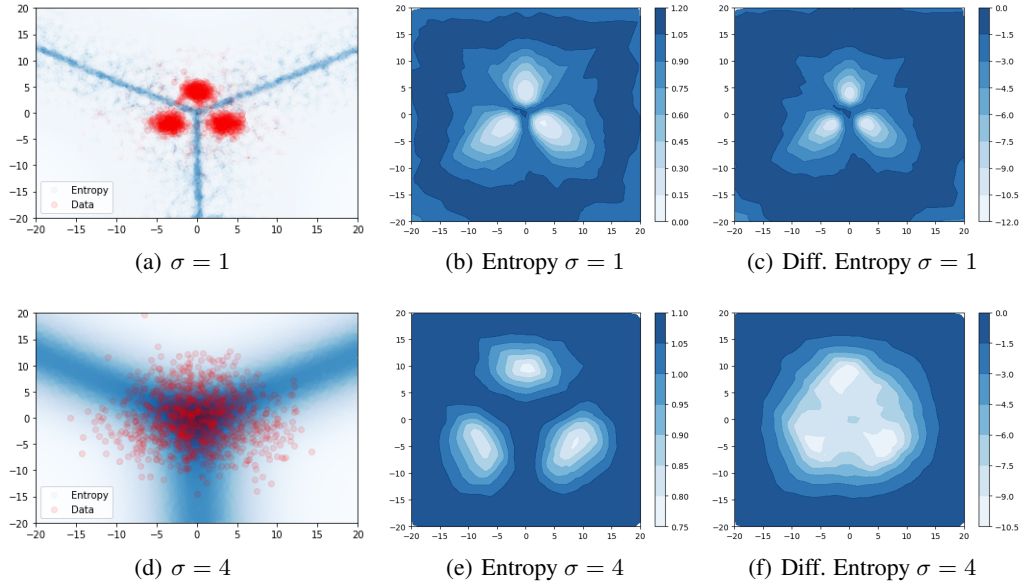

(a) $\sigma = 1$  (b) Entropy $\sigma = 1$  (c) Diff. Entropy $\sigma = 1$

(d) $\sigma = 4$  (e) Entropy $\sigma = 4$  (f) Diff. Entropy $\sigma = 4$

Figure 3: Synthetic Experiment

is out-of-distribution given a measure of uncertainty. Out-of-distribution samples are chosen as the positive class. The OMNIGLOT dataset [37], scaled down to 28x28 pixels, was used as real 'OOD' data for MNIST. 15000 samples of OMNIGLOT data were randomly selected to form a balanced set of positive (OMNIGLOT) and negative (MNIST valid+test) samples. For CIFAR-10 three OOD datasets were considered - SVHN, LSUN and TinyImagetNet (TIM) [38, 39, 40]. The two considered baseline approaches derive uncertainty measures from either the class posterior of a DNN [13] or an ensemble generated via MC dropout applied to the same DNN [23, 18]. All uncertainty measures described in section 4 are explored for both tasks in order to see which yield best performance. The performance is assessed by area under the ROC (AUROC) and Precision-Recall (AUPR) curves in both experiments as in [13].

Table 1: MNIST and CIFAR-10 misclassification detection

| Data | Model | AUROC | | | | AUPR | | | | % Err. |
|------|-------|-------|------|------|--------|-------|------|------|--------|--------|
| | | Max.P | Ent. | M.I. | D.Ent. | Max.P | Ent. | M.I. | D.Ent. | |
| MNIST | DNN | 98.0 | 98.6 | - | - | 26.6 | 25.0 | - | - | **0.4** |
| | MCDP | 97.2 | 97.2 | 96.9 | - | 33.0 | 29.0 | 27.8 | - | **0.4** |
| | DPN | **99.0** | 98.9 | 98.6 | 92.9 | **43.6** | 39.7 | 30.7 | 25.5 | 0.6 |
| CIFAR10 | DNN | 92.4 | 92.3 | - | - | 48.7 | 47.1 | - | - | **8.0** |
| | MCDP | **92.5** | 92.0 | 90.4 | - | 48.4 | 45.5 | 37.6 | - | **8.0** |
| | DPN | 92.2 | 92.1 | 92.1 | 90.9 | **52.7** | **51.0** | **51.0** | 45.5 | 8.5 |

Table 1 shows that the DPN consistently outperforms both a DNN, and a MC dropout ensemble (MCDP) in misclassification detection performance, although there is a negligible drop in accuracy of the DPN as compared to a DNN or MCDP. Max probability yields the best results, closely followed by the entropy of the predictive distribution. This is expected, as they are measures of total uncertainty in predictions, while the other measures capture the either *model* or *distributional uncertainty*. The performance difference is more pronounced on AUPR, which is sensitive to misbalanced classes.

Table 2 shows that a DPN consistently outperforms the baselines in OOD sample detection for both MNIST and CIFAR-10 datasets. On MNIST, the DPN is able to perfectly classify all samples using max probability, entropy and differential entropy. On the CIFAR-10 dataset the DPN consistently outperforms the baselines by a large margin. While high performance against SVHN and LSUN is expected, as LSUN, SVHN and CIFAR-10 are quite different, high performance against TinyImageNet, which is also a dataset of real objects and therefore closer to CIFAR-10, is more impressive. Curiously, MC dropout does not always yield better results than a standard DNN, which supports

the assertion that it is difficult to achieve the desired behaviour for a Bayesian distribution over distributions.

Table 2: MNIST and CIFAR-10 out-of-domain detection

| Data | | Model | AUROC | | | | AUPR | | | |
| ID | OOD | | Max.P | Ent. | M.I. | D.Ent. | Max.P | Ent. | M.I. | D.Ent. |
|---|---|---|---|---|---|---|---|---|---|---|
| MNIST | OMNI | DNN | 98.7 | 98.8 | - | - | 98.3 | 98.5 | - | - |
| | | MCDP | 99.2 | 99.2 | 99.3 | - | 99.0 | 99.1 | 99.3 | - |
| | | DPN | **100.0** | **100.0** | 99.5 | **100.0** | **100.0** | **100.0** | 97.5 | **100.0** |
| CIFAR10 | SVHN | DNN | 90.1 | 90.8 | - | - | 84.6 | 85.1 | - | - |
| | | MCDP | 89.6 | 90.6 | 83.7 | - | 84.1 | 84.8 | 73.1 | - |
| | | PN | 98.1 | 98.2 | 98.2 | **98.5** | 97.7 | 97.8 | 97.8 | **98.2** |
| CIFAR10 | LSUN | DNN | 89.8 | 91.4 | - | - | 87.0 | 90.0 | - | - |
| | | MCDP | 89.1 | 90.9 | 89.3 | - | 86.5 | 89.6 | 86.4 | - |
| | | DPN | 94.4 | 94.4 | 94.4 | **94.6** | 93.3 | **93.4** | **93.4** | 93.3 |
| CIFAR10 | TIM | DNN | 87.5 | 88.7 | - | - | 84.7 | 87.2 | - | - |
| | | MCDP | 87.6 | 89.2 | 86.9 | - | 85.1 | 87.9 | 83.2 | - |
| | | DPN | 94.3 | 94.3 | 94.3 | **94.6** | 94.0 | 94.0 | 94.0 | **94.2** |

The experiments above suggest that there is little benefit of using measures such as differential entropy and mutual information over standard entropy. However, this is because MNIST and CIFAR-10 are low data uncertainty datasets - all classes are distinct. It is interesting to see whether differential entropy of the Dirichlet prior will be able to distinguish in-domain and out-of-distribution data better than entropy when the classes are less distinct. To this end zero mean isotropic Gaussian noise with a standard deviation $\sigma = 3$ noise is added to the inputs of the DNN and DPN during both training and evaluation on the MNIST dataset. Table 3 shows that in the presence of strong noise entropy and MI fail to successfully discriminate between in-domain and out-of-distribution samples, while performance using differential entropy barely falls.

Table 3: MNIST vs OMNIGLOT. Out-of-distribution detection AUROC on noisy data.

| | Ent. | | M.I. | | D.Ent. | |
| $\sigma$ | 0.0 | 3.0 | 0.0 | 3.0 | 0.0 | 3.0 |
|---|---|---|---|---|---|---|
| DNN | 98.8 | 58.4 | - | - | - | - |
| MCDP | 98.8 | 58.4 | 99.3 | 79.1 | - | - |
| DPN | 100.0 | 51.8 | 99.5 | 22.3 | 100.0 | 99.8 |

## 6 Conclusion

This work describes the limitations of previous work on predictive uncertainty estimations within the context of sources of uncertainty and proposes to treat out-of-distribution (OOD) inputs as a separate source of uncertainty, called *Distributional Uncertainty*. To this end, this work presents a novel framework, called Prior Networks (PN), which allows *data, distributional* and *model uncertainty* to be treated separately within a consistent probabilistically interpretable framework. A particular form of these PNs are applied to classification, Dirichlet Prior Networks (DPNs). DPNs are shown to yield more accurate estimates of distributional uncertainty than MC Dropout and standard DNNs on the task of OOD detection on the MNIST and CIFAR-10 datasets. The DPNs also outperform other methods on the task of misclassification detection. A range of uncertainty measures is presented and analyzed in the context of the types of uncertainty which they assess. It was noted that measures of total uncertainty, such as max probability or entropy of the predictive distribution, yield the best results on misclassification detection. Differential entropy of DPN was best for measure of uncertainty for OOD detection, especially when classes are less distinct. This was illustrated on both a synthetic experiment and on a noise-corrupted MNIST task. Uncertainty measures can be analytically calculated at test time for DPNs, reducing computational cost relative to ensemble approaches. Having investigated PNs for image classification, it is interesting to apply them to other tasks computer vision, NLP, machine translation, speech recognition and reinforcement learning. Finally, it is necessary to explore Prior Networks for regression tasks.

**Acknowledgments**

This paper reports on research partly supported by Cambridge Assessment, University of Cambridge. This work also partly funded by a DTA EPSRC away and a Google Research award. We would also like to thank members of the CUED Machine Learning group, especially Dr. Richard Turner, for fruitful discussions.

## Footnotes

[1] Up to identifiability limits. In the limit of infinite data $\mathrm{p}(\boldsymbol{\theta} | \mathcal{D})$ yields equivalent parameterizations.

[2] Code available at https://github.com/KaosEngineer/DirichletPriorNetworks

[3] Using the standard shorthand for $\mathrm{P}(y = \omega_c | \boldsymbol{x}^*, \mathcal{D})$.

[4]Where $\boldsymbol{\mu}$ is a vector of probabilities: $\begin{bmatrix} \mu_1, & \cdots, & \mu_K \end{bmatrix}^T = \begin{bmatrix} \mathrm{P}(y = \omega_1), & \cdots, & \mathrm{P}(y = \omega_K) \end{bmatrix}^T$

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
