[Supplementary Material]

# Appendix A    Experimental Setup and Datasets

For both core and additional experiments models were trained on the MNIST [35], SVHN [38] and CIFAR [36] datasets. Dataset sizes can be found in table 4. In addition to the datasets described

Table 4: Training and Evaluation Datasets

| Dataset | Train | Valid | Test | Classes |
|---------|-------|-------|------|---------|
| MNIST | 55000 | 5000 | 10000 | |
| SVHN | 73257 | - | 26032 | 10 |
| CIFAR-10 | 50000 | - | 10000 | |
| CIFAR-100 | 50000 | - | 10000 | 100 |

above, the OMNIGLOT [37], SEMEION [41], LSUN [39] and TinyImagenet [40] datasets were used for out-of-distribution input detection experiments. For these datasets only their test sets were used, described in table 4. TinyImagenet was resized down to 32x32 from 64x64 and OMNIGLOT was resized down to 28x28 using bilinear interpolation. For all datasets the input features were re-scaled

Table 5: Additional Evaluation Datasets

| Dataset | Size |
|---------|------|
| OMNIGLOT | 32460 |
| SEMEION | 1593 |
| LSUN | 10000 |
| tinyImagenet | 10000 |

to the range -1.0 and 1.0 from the range 0 and 255. No additional preprocessing was done models trained on the MNIST and SVHN datasets. For models trained on CIFAR-10, images were randomly flipped left-right, shifted by $\pm 4$ pixels and rotated by $\pm 15$ degrees as a form of data augmentation.

All networks for all experiments were constructed using variants on the VGG [2] architecture for image classification. Models were implemented in Tensorflow [42]. Details of the architectures used for each dataset can be found in table 6. For convolutional layers dropout was used with a higher keep probability than for fully-connected layers.

Table 6: Architecture Sizes

| Dataset | Arch. | Activation | Conv Depth | FC Layers | FC units |
|---------|-------|------------|------------|-----------|----------|
| MNIST | VGG-6 | ReLU | 4 | 1 | 100 |
| SVHN | VGG-16 | Leaky ReLU | 13 | 2 | 2048 |
| CIFAR-10 | VGG-16 | Leaky ReLU | 13 | 2 | 2048 |

The training configuration for all models is described in table 7. Interestingly, it was necessary to use less dropout for the DPN, due to the regularization effect of the noise data. All models trained using the NADAM optimizer [43]. For the models trained on MNIST exponentially decaying learning rates were used. Models trained on SVHN and CIFAR-10 used 1-Cycle learning rates, where learning rates are linearly increased from the initial learning rate to 10x the initial learning rate for half a cycle and then linearly decreased back down to the initial learning rate for the remained of the cycle. Learning rates are then linearly decreased until 1e-6 for the remaining training epochs. This approach has been shown to act both as a regularizer as well as speed up training of models [?].

Table 7: Training Configuration

| Dataset | Model | Dropout | LR | Cycle Len. | Epochs | $\hat{\alpha}_0$ | CE weight | OOD data |
|---------|-------|---------|------|------------|--------|------------------|-----------|----------|
| MNIST | DNN | 0.50 | 1e-3 | - | 30 | - | - | - |
| | DPN | 0.95 | 1e-3 | - | 10 | 1e3 | 0.0 | MNIST FA |
| SVHN | DNN | 0.50 | 1e-3 | 30 | 40 | - | - | - |
| | DPN | 0.50 | 7.5e-4 | 30 | 40 | 1e3 | 1.0 | CIFAR-10 |
| CIFAR-10 | DNN | 0.50 | 1e-3 | 30 | 45 | - | - | - |
| | DPN | 0.70 | 7.5e-4 | 70 | 100 | 1e2 | 1.0 | CIFAR-100 |

For the DPN trained on MNIST data the out-of-distribution data was synthesized using a Factor Analysis model with a 50-dimensional latent space. In standard factor analysis the latent vectors have an isotropic standard normal distribution. To push the FA model to produce data at the boundary of the in-domain region the variance on the latent distribution was increased.

## Appendix B   Additional Experiments

Further experiments have been run in addition to the core experiments described in section 5. In appendix B.1 the MNIST DNN and DPN described in section 5.2 is evaluated against other out-of-distribution datasets. In appendix B.2 and B.3 a DPN is trained on the SVHN [38] and CIFAR-10 [36] datasets, respectively, and evaluated on the tasks of misclassification detection and out-of-distribution input detection.

### B.1   Additional MNIST experiments

In Table 8 out-of-distribution input detection is run against the SEMEION, SVHN and CIFAR-10 datasets. SEMEION is a dataset of greyscale handwritten 16x16 digits, whose primary difference from MNIST is that there is no padding between the edge of the image and the digit. SEMEION digits were upscaled to 28x28 for these experiments. For the SVHN and CIFAR-10 experiments, the images were transformed into greyscale and downsampled to 28x28 size.

The purpose here is to investigate how out-of-distribution input detection performance is affected by the similarity of the OOD data to the in-domain data. Here, SEMEION is the most similar dataset to MNIST, as it is also composed of greyscale handwritten digits. SVHN, also a dataset over digits 0-9, is less similar, as the digits are now embedded in street signs. CIFAR-10 is the most different, as it is a dataset of real objects. In all experiments presented in table 8 the DPN outperforms the baselines. Performance of all models is worst on SEMEION and best on CIFAR-10, illustrating how OOD detection is more challenging as the datasets become less distinct. Note, As SEMEION is a very small dataset it was not possible to get a balanced set of MNIST and SEMEION images, so AUPR is a better performance metric than AUROC on this particular experiment.

Table 8: MNIST out-of-domain detection

| OOD Data | Model | AUROC | | | | AUPR | | | |
|---|---|---|---|---|---|---|---|---|---|
| | | Max.P | Ent. | M.I. | D.Ent. | Max.P | Ent. | M.I. | D.Ent. |
| SEMEION | DNN | 92.7 | 92.9 | - | - | 76.4 | 76.7 | - | - |
| | MCDP | 95.2 | 95.3 | 95.4 | - | 84.1 | 84.2 | 87.3 | - |
| | DPN | 99.5 | 99.6 | 99.1 | **99.7** | 96.9 | 97.5 | 90.8 | **98.6** |
| SVHN | DNN | 98.7 | 98.9 | - | - | 98.5 | 98.7 | - | - |
| | MCDP | 98.2 | 98.4 | 98.1 | - | 98.0 | 98.3 | 97.9 | - |
| | DPN | 99.9 | **100.0** | 99.5 | **100.0** | 99.9 | **100.0** | 98.5 | **100.0** |
| CIFAR10 | DNN | 99.4 | 99.5 | - | - | 99.3 | 99.4 | - | - |
| | MCDP | 99.1 | 99.3 | 98.9 | - | 98.9 | 99.2 | 98.6 | - |
| | DPN | **100.0** | **100.0** | 99.5 | **100.0** | **100.0** | **100.0** | 98.2 | **100.0** |

### B.2   SVHN Experiments

This section describes misclassification and out-of-distribution input detection experiments on the SVHN dataset. A DPN trained on SVHN used the CIFAR-10 dataset as the noise dataset, rather than using a generative model like Factor Analysis, VAE or GAN. Investigation of appropriate methods to synthesize out-of-distribution data for complex datasets is beyond the scope of this work.

Table 9 describes the misclassification detection experiment on SVHN. Note, all models achieve comparable classification error (4.3-5.1%). The DPN outperforms the baselines according to AUPR but achieves lower performance in AUROC on misclassification detection using all measures.

Table 10 reports the out-of-distribution detection performance of SVHN vs CIFAR-10, CIFAR-100, LSUN and TinyImageNet datasets, respectively. In all experiments the DPN is seen to consistently achieves highest performance. Note, the DPN uses CIFAR-10 as the training out-of-distribution

Table 9: SVHN test misclassification detection

| Model | AUROC | | | | AUPR | | | | % Err. |
|-------|-------|------|------|-------|-------|------|------|-------|--------|
| | Max.P | Ent. | M.I. | D.Ent. | Max.P | Ent. | M.I. | D.Ent. | |
| DNN | 90.1 | 91.8 | - | - | 47.7 | 46.8 | - | - | **4.3** |
| MCDP | 92.0 | **92.2** | 92.0 | - | 46.4 | 43.5 | 40.4 | - | **4.3** |
| DPN | 90.1 | 90.1 | 90.1 | 91.2 | **55.3** | 54.8 | 54.8 | 46.0 | 5.1 |

dataset, so it is unsurprising that it achieves near-perfect performance on a held-out set of CIFAR-10 data. Interestingly, there is a larger margin between the DNN and MCDP on SVHN than on networks trained either on MNIST or CIFAR-10.

Table 10: SVHN out-of-domain detection

| OOD Data | Model | AUROC | | | | AUPR | | | |
|----------|-------|-------|------|------|-------|-------|------|------|-------|
| | | Max.P | Ent. | M.I. | D.Ent. | Max.P | Ent. | M.I. | D.Ent. |
| CIFAR10 | DNN | 92.5 | 93.8 | - | - | 91.4 | 92.1 | - | - |
| | MCDP | 95.6 | 96.0 | 96.3 | - | 94.4 | 95.0 | 95.8 | - |
| | DPN | **99.9** | **99.9** | **99.9** | **99.9** | **100.0** | **100.0** | **100.0** | 99.9 |
| CIFAR100 | DNN | 92.4 | 93.8 | - | - | 91.4 | 92.1 | - | - |
| | MCDP | 94.2 | 94.8 | 95.4 | - | 94.2 | 94.8 | 95.4 | - |
| | DPN | **99.8** | **99.8** | **99.8** | **99.8** | **99.8** | **99.8** | **99.8** | **99.8** |
| LSUN | DNN | 91.9 | 93.4 | - | - | 90.7 | 91.3 | - | - |
| | MCDP | 95.9 | 96.3 | 97.0 | - | 94.9 | 95.3 | 96.8 | - |
| | DPN | **100.0** | **100.0** | **100.0** | **100.0** | 99.9 | 99.9 | 99.9 | **100.0** |
| TIM | DNN | 93.1 | 94.2 | - | - | 91.8 | 92.5 | - | - |
| | MCDP | 96.3 | 96.7 | 97.1 | - | 95.3 | 95.8 | 96.8 | - |
| | DPN | **100.0** | **100.0** | **100.0** | **100.0** | 99.9 | 99.9 | 99.9 | **100.0** |

## B.3 CIFAR-10 Experiments

This section presents the results of misclassification and out-of-distribution input detection experiments on the CIFAR-10 dataset. A DPN trained on CIFAR-10 used the CIFAR-100 dataset as the out-of-distribution training dataset. CIFAR-100 is similar to CIFAR-10 but describes different objects than CIFAR-10, so there is no class overlap. This is the most challenging set of experiments, as visually CIFAR-10 is much more similar to CIFAR-100, LSUN and TinyImageNet, so out-of-distribution input detection is likely to more difficult than for simpler tasks like MNIST and SVHN.

Table 11 gives the results of the misclassification detection experiment on CIFAR-10. All models achieve comparable classification error (8-8.5%), with the DPN achieving a slightly higher performance than the baselines in AUPR.

Table 11: CIFAR-10 test misclassification detection

| Model | AUROC | | | | AUPR | | | | % Err. |
|-------|-------|------|------|-------|-------|------|------|-------|--------|
| | Max.P | Ent. | M.I. | D.Ent. | Max.P | Ent. | M.I. | D.Ent. | |
| DNN | 92.4 | 92.3 | - | - | 48.7 | 47.1 | - | - | **8.0** |
| MCDP | **92.5** | 92.0 | 90.4 | - | 48.4 | 45.5 | 37.6 | - | **8.0** |
| DPN | 92.2 | 92.1 | 92.1 | 90.9 | **52.7** | 51.0 | 51.0 | 45.5 | 8.5 |

Table 12 reports the results of the out-of-distribution detection of CIFAR-10 vs CIFAR-100, SVHN, LSUN and TinyImageNet datasets. In all experiments the DPNs achieve the best performance, outperforming the baselines by a larger margin than previously. Note, CIFAR-100 is used as OOD training data for the DPN, so high performance on it is expected. TinyImageNet is the most similar to CIFAR-10 (other than CIFAR-100) and it the most challenging OOD detection task, as the baseline approaches achieve the lowest performance on it. Notably, In each experiment the performance of the baseline approaches is noticeable lower than before, especially using mutual information of MCDP as a measure of uncertainty. This indicates that it is indeed difficult to control the behaviour of Bayesian distributions over distributions for complex tasks. This set of experiments clearly demonstrates that

Prior Networks perform well on much more difficult datasets than MNIST and are able to outperform previously proposed Bayesian and non-Bayesian approaches.

Table 12: CIFAR-10 out-of-domain detection

| OOD Data | Model | AUROC | | | | AUPR | | | |
|---|---|---|---|---|---|---|---|---|---|
| | | Max.P | Ent. | M.I. | D.Ent. | Max.P | Ent. | M.I. | D.Ent. |
| CIFAR100 | DNN | 86.4 | 87.2 | - | - | 82.6 | 84.3 | - | - |
| | MCDP | 86.4 | 87.5 | 85.7 | - | 83.0 | 84.9 | 81.5 | - |
| | DPN | 95.6 | 95.7 | 95.7 | **95.8** | 95.1 | 95.1 | 95.1 | **95.5** |
| SVHN | DNN | 90.1 | 90.8 | - | - | 84.6 | 85.1 | - | - |
| | MCDP | 89.6 | 90.6 | 83.7 | - | 84.1 | 84.8 | 73.1 | - |
| | DPN | 98.1 | 98.2 | 98.2 | **98.5** | 97.7 | 97.8 | 97.8 | **98.2** |
| LSUN | DNN | 89.8 | 91.4 | - | - | 87.0 | 90.0 | - | - |
| | MCDP | 89.1 | 90.9 | 89.3 | - | 86.5 | 89.6 | 86.4 | - |
| | DPN | 94.4 | 94.4 | 94.4 | **94.6** | 93.3 | **93.4** | **93.4** | 93.3 |
| TIM | DNN | 87.5 | 88.7 | - | - | 84.7 | 87.2 | - | - |
| | MCDP | 87.6 | 89.2 | 86.9 | - | 85.1 | 87.9 | 83.2 | - |
| | DPN | 94.3 | 94.3 | 94.3 | **94.6** | 94.0 | 94.0 | 94.0 | **94.2** |

## Appendix C   Derivations for Uncertainty Measures and KL divergence

This appendix provides the derivations and shows how calculate the uncertainty measures discussed in section 4 for a DNN/DPN and a Bayesian Monte-Carlo Ensemble. Additionally, it describes how to calculate the KL divergence between two Dirichlet distributions.

### C.1   Entropy of Predictive Distribution for Bayesian MC Ensemble

Entropy of the predictive posterior can be calculated for a Bayesian MC Ensemble using the following derivation, which is taken from Yarin Gal's PhD thesis [23].

$$
\begin{aligned}
\mathcal{H}[\mathrm{P}(y|\boldsymbol{x}^*, \mathcal{D})] &= -\sum_{c=1}^{K} \mathrm{P}(\omega_c|\boldsymbol{x}^*, \mathcal{D}) \ln \mathrm{P}(\omega_c|\boldsymbol{x}^*, \mathcal{D}) \\
&= -\sum_{c=1}^{K} \left( \int \mathrm{p}(\omega_c|\boldsymbol{x}^*, \boldsymbol{\theta}) \mathrm{p}(\boldsymbol{\theta}|\mathcal{D}) d\boldsymbol{\theta} \right) \ln \left( \int \mathrm{P}(\omega_c|\boldsymbol{x}^*, \boldsymbol{\theta}) \mathrm{p}(\boldsymbol{\theta}|\mathcal{D}) d\boldsymbol{\theta} \right) \\
&\approx -\sum_{c=1}^{K} \left( \int \mathrm{P}(\omega_c|\boldsymbol{x}^*, \boldsymbol{\theta}) \mathrm{q}(\boldsymbol{\theta}) d\boldsymbol{\theta} \right) \ln \left( \int \mathrm{P}(\omega_c|\boldsymbol{x}^*, \boldsymbol{\theta}) \mathrm{q}(\boldsymbol{\theta}) d\boldsymbol{\theta} \right) \\
&\approx -\sum_{c=1}^{K} \left( \frac{1}{M} \sum_{i=1}^{M} \mathrm{P}(\omega_c|\boldsymbol{x}^*, \boldsymbol{\theta}^{(i)}) \right) \ln \left( \frac{1}{M} \sum_{i=1}^{M} \mathrm{P}(\omega_c|\boldsymbol{x}^*, \boldsymbol{\theta}^{(i))}) \right)
\end{aligned}
$$

### C.2   Differential Entropy of Dirichlet Prior Network

The derivation of differential entropy simply quotes the standard result for Dirichlet distributions. Notably the $\alpha_c$ are a function of $\boldsymbol{x}^*$ and $\psi$ is the *digamma function* and $Gamma$ is the *Gamma function*.

$$
\begin{aligned}
\mathcal{H}[\mathrm{p}(\boldsymbol{\mu}|\boldsymbol{x}^*; \hat{\boldsymbol{\theta}})] &= -\int_{\mathcal{S}^{K-1}} \mathrm{p}(\boldsymbol{\mu}|\boldsymbol{x}; \hat{\boldsymbol{\theta}}) \ln(\mathrm{p}(\boldsymbol{\mu}|\boldsymbol{x}; \hat{\boldsymbol{\theta}})) d\boldsymbol{\mu} \\
&= \sum_{c}^{K} \ln \Gamma(\alpha_c) - \ln \Gamma(\alpha_0) - \sum_{c}^{K} (\alpha_c - 1) \cdot (\psi(\alpha_c) - \psi(\alpha_0))
\end{aligned}
$$

### C.3 Mutual Information for Bayesian MC Ensemble

The Mutual information between class label and parameters can be calculated for a Bayesian MC Ensemble using the following derivation, which is also taken from Yarin Gal's PhD thesis [23]:

$$
\underbrace{\mathcal{I}[y,\boldsymbol{\theta}|\boldsymbol{x}^*,\mathcal{D}]}_{Model\ Uncertainty} = \underbrace{\mathcal{H}[\mathbb{E}_{\mathrm{p}(\boldsymbol{\theta}|\mathcal{D})}[\mathrm{P}(y|\boldsymbol{x}^*,\boldsymbol{\theta})]]}_{Total\ Uncertainty} - \underbrace{\mathbb{E}_{\mathrm{p}(\boldsymbol{\theta}|\mathcal{D})}[\mathcal{H}[\mathrm{P}(y|\boldsymbol{x}^*,\boldsymbol{\theta})]]}_{Expected\ Data\ Uncertainty}
$$

$$
\approx \mathcal{H}[\mathbb{E}_{\mathrm{q}_\theta(\boldsymbol{\theta})}[\mathrm{P}(y|\boldsymbol{x}^*,\boldsymbol{\theta})]] - \mathbb{E}_{\mathrm{q}_\theta(\boldsymbol{\theta})}[\mathcal{H}[\mathrm{P}(y|\boldsymbol{x}^*,\boldsymbol{\theta})]]
$$

$$
\approx \mathcal{H}[\frac{1}{M}\sum_{i=1}^{M}\mathrm{P}(\omega_c|\boldsymbol{x}^*,\boldsymbol{\theta}^{(i)})] - \frac{1}{M}\sum_{i=1}^{M}\mathcal{H}[\mathrm{P}(y|\boldsymbol{x}^*,\boldsymbol{\theta}^{(i)})]
$$

### C.4 Mutual Information for Dirichlet Prior Network

The mutual information between the labels y and the categorical $\boldsymbol{\mu}$ for a DPN can be calculated as follows, using the fact that MI is the difference of the entropy of the expected distribution and the expected entropy of the distribution.

$$
\underbrace{\mathcal{I}[y,\boldsymbol{\mu}|\boldsymbol{x}^*,\hat{\boldsymbol{\theta}}]}_{Distributional\ Uncertainty} = \underbrace{\mathcal{H}[\mathbb{E}_{\mathrm{p}(\boldsymbol{\mu}|\boldsymbol{x}^*,\hat{\boldsymbol{\theta}})}[\mathrm{P}(y|\boldsymbol{\mu})]]}_{Total\ Uncertainty} - \underbrace{\mathbb{E}_{\mathrm{p}(\boldsymbol{\mu}|\boldsymbol{x}^*,\hat{\boldsymbol{\theta}})}[\mathcal{H}[\mathrm{P}(y|\boldsymbol{\mu})]]}_{Expected\ Data\ Uncertainty}
$$

$$
= \mathcal{H}[\mathrm{P}(y|\boldsymbol{x}^*,\hat{\boldsymbol{\theta}})] + \sum_{c=1}^{K}\mathbb{E}_{\mathrm{p}(\boldsymbol{\mu}|\boldsymbol{x}^*,\hat{\boldsymbol{\theta}})}[\mu_c\ln\mu_c]
$$

$$
= -\sum_{c=1}^{K}\frac{\alpha_c}{\alpha_0}\left(\ln\frac{\alpha_c}{\alpha_0} - \psi(\alpha_c+1) + \psi(\alpha_0+1)\right)
$$

The second term in this derivation is a non-standard result. The expected entropy of the distribution can be calculated in the following way:

$$
\mathbb{E}_{\mathrm{p}(\boldsymbol{\mu}|\boldsymbol{x}^*,\hat{\boldsymbol{\theta}})}[\mu_c\ln(\mu_c)] = \frac{\Gamma(\alpha_0)}{\prod_{c=1}^{K}\Gamma(\alpha_c)}\int_{\mathcal{S}_K}\mu_c\ln(\mu_c)\prod_{c=1}^{K}\mu_c^{\alpha_c-1}d\boldsymbol{\mu}
$$

$$
= \frac{\alpha_c}{\alpha_0}\frac{\Gamma(\alpha_0+1)}{\Gamma(\alpha_c+1)\prod_{c'=1,\neq c}^{K}\Gamma(\alpha_{c'})}\int_{\mathcal{S}_K}\mu_c^{\alpha_c}\ln(\mu_c)\prod_{c'=1,\neq c}^{K}\mu_{c'}^{\alpha_{c'}-1}d\boldsymbol{\mu}
$$

$$
= \frac{\alpha_c}{\alpha_0}(\psi(\alpha_c+1) - \psi(\alpha_0+1))
$$

Here the expectation is calculated by noting that the standard result of the expectation of $\ln\mu_c$ wrt a Dirichlet distribution can be used if the extra factor $\mu_c$ is accounted for by adding 1 to the associated concentration parameter $\alpha_c$ and multiplying by $\frac{\alpha_c}{\alpha_0}$ in order to have the correct normalizing constant.

### C.5 KL Divergence between two Dirichlet Distributions

The KL divergence between two Dirichlet distributions $\mathrm{p}(\boldsymbol{\mu}|\boldsymbol{\alpha})$ and $\mathrm{p}(\boldsymbol{\mu}|\boldsymbol{\beta})$ can be obtained in closed form as follows:

$$
KL[\mathrm{p}(\boldsymbol{\mu}|\boldsymbol{\alpha})||\mathrm{p}(\boldsymbol{\mu}|\boldsymbol{\beta})] = \mathbb{E}_{\mathrm{p}(\boldsymbol{\mu}|\boldsymbol{\alpha})}[\ln\mathrm{p}(\boldsymbol{\mu}|\boldsymbol{\alpha}) - \ln\mathrm{p}(\boldsymbol{\mu}|\boldsymbol{\beta})]
$$

$$
= \ln\Gamma(\alpha_0) - \ln\Gamma(\beta_0) + \sum_{c=1}^{K}\ln\Gamma(\beta_c) - \ln\Gamma(\alpha_c)
$$

$$
+ \sum_{c=1}^{K}(\alpha_c - \beta_c)\mathbb{E}_{\mathrm{p}(\boldsymbol{\mu}|\boldsymbol{\alpha})}[\ln(\mu_c)]
$$

$$
= \ln\Gamma(\alpha_0) - \ln\Gamma(\beta_0) + \sum_{c=1}^{K}\ln\Gamma(\beta_c) - \ln\Gamma(\alpha_c)
$$

$$
+ \sum_{c=1}^{K}(\alpha_c - \beta_c)(\psi(\alpha_c) - \psi(\alpha_0))
$$