[Reviews · NeurIPS 2018]

Reviewer 1



The authors develop prior networks, an approach to identifying misclassification on in and out of distribution examples. The work largely builds upon Malinin et al, 2017, adapting their method to the Dirichlet case. The experiments are simple but lack controls and may be too simple to validate the method. Overall: Quality: There do not appear to be any obvious errors. There is no model uncertainty in this paper as a dirac is used for the model prior, so why spend time talkimg about it? Clarity: The presentation at times seems convoluted and several confusing terms are introduced for apparently the same thing, even where standard terms exist for these in statistics. Some of the explanation is unclear and needs refinement. Originality: It is a fairly straightforward adaptation of Malinin et al, 2017 to the Dirichlet case (using a net for the concentration parameter instead of the variance). Significance: The experimental evaluation is weak. (See comments below) Comments: Intro: The epistemic/aleatoric uncertainty terms are not useful. It has been related to the explained/unexplained variance in the law of total variance, which differs from reducible/irreducible uncertainty in general. Model/data/distributional uncertainty are much clearer notions, are well defined, and are analogous to terms in statistical learning theory. It's worth noting that variantional methods have also been applied to (1), most recently see Graves, 2011, and Blundell et al, 2015, Kingma et al, 2015, Louizos and Welling, 2016. Eq (2)/(3): this explanation needs improving. 1) Instead of P(y=w_c), why not use capital letters for the random variables and lower case (with subscripts if you like) for their values? 2) eq (3) should include the conditional for each P(y = w_c) term: they are not marginals, it's just a vector of the terms in the sum of eq (2). Line 94 talks about decision boundaries, but there are not any defined in text or the figures (you need to specify a loss over choices to identify the boundary). I'm also not sure aim to have increasingly varied decision boundaries far from training data, how is this justified? Section 3: it seems sensible to mention that what is proposed is a hierarchical Bayesian model, in the sense there are now three layers: the likelihood, the per-data prior distribution and the global prior distribution. This idea has been extensively explored in non-neural bayesian models (e.g., LDA). (7) and (8) were a surprise. They effectively eliminate the whole hierarchy that was introduced in Section 3. It makes the whole story about Bayesian inference a bit overboard: in fact, there is very little to do with Bayes here. Indeed, it's not stated explicitly that I could see, but p(w_c|mu) = mu_c and so, fact, everything simplifies to eq (12) which shows that essentially the prediction is just a softmax over a neural network applied to the inputs in eq (13). Whereas Malinin et al, 2017 use (14) to fit Gaussians, this work use it to fit categorical data. That appears to be the main distinction to prior work. This work gives no further justification to this heuristic (they appear to set the coefficient trading off between these two tasks to one without justification). Certainly it looks like parameter estimation by this approach will be biased, but what other properties does it possess? The experiments consist of an experiment on a toy Gaussian discrimination task and on MNIST vs OMNIGLOT out of distribution discrimination. It's not unremarkable that PN worked on the toy task (but it is not really a strong scientific test; there's no baseline, maybe the task is too easy). The MNIST vs OMNIGLOT tasks is concerning. Performance of the baseline models is already high on these tasks, and no error bars are included, hence we do not know if the task is simply too easy and there is no difference in performance between the proposal and other obvious approaches.

Reviewer 2



The paper describe a novel interpretable framework to measure the uncertainty of a prediction from NNs. The paper is clear and very well-written. Besides, lots of intuitions behind are described, which is very helpful to understand the paper. The paper is also original since it is quite different from previous Bayesian approach by explicitly parametrize the distribution by DNN. However I have some concerns over significance of the paper. The results are very well-presented as well by visualization (Fig2) or by AUROC/AUPR. The data used in paper, on the other hand, is quite trivial. More results(say in CIFAR 10) will definitely make the paper more stronger. More questions, Q1: should x in line 82 be x^* in P(\omega_c| x, \theta)? Q2: Cannot understand the last part of the sentence at line 92 to line 95 Q3: notation \prod_c^K or \sum_c^K is a little bit confusing, to me the common uses are either \sum_{c=1}^K or \sum_{c\in [K]} where [K] is a set.

Reviewer 3



This paper introduces Prior networks which help distinguish uncertainty in posterior estimates due distributional uncertainty and data uncertainty. This is achieved by introducing an intermediate distribution to represent the distributional uncertainty. This probability distribution is parametrized and the parameters are estimated using a neural network. More specifically the paper discusses this parametrization using Dirichlet distribution and estimates the concentration parameters using the neural network. It uses a max-entropy inspired KL divergence criterion to train the network to distinguish in-domain and out-of-domain samples. The Dirichlet distribution corresponding to the in-network distribution is re-parametrized to simplify specification. This is a very well written paper with a thorough survey of existing approaches, providing a valuable reading list. Further it provides a detailed introduction to the problem. The presentation is very clear and the authors clearly distinguish their contributions from previous work. A major assumption in this class of works is that "out-of-distribution" samples are specified. However the definition of out-of-distribution is usually not very clear and can be application dependent. Further the OOD detection performance can significantly vary depending on what is considered OOD e.g. in speech recognition applications out-of-language, out-of-vocabulary or unknown noise conditions can all be considered OOD depending on the application and as shown in [1] the OOD detection performance significantly varies according to this definition. It would be informative to see how the proposed method performs depending on this confusability between in-domain and out-of-domain detection compared to baseline methods. Providing a more elaborate description of the training method in section 3.2 would be very informative. Choosing a different colormap in figures 2{b,c,e,f} and providing a colorbar would be helpful. "model uncertainty" is used with two different meanings in this paper and it can be confusing as to what it means without careful reading. This even happens in the abstract. I would recommend italicizing the noun to avoid confusion. [1] Dan Hendrycks and Kevin Gimpel, “A Baseline for Detecting Misclassified and Out-of- Distribution Examples in Neural Networks,” http://arxiv.org/abs/1610.02136, 2016, arXiv:1610.02136.